# Timing of the Pubertal Growth Spurt and Prostate Cancer

**DOI:** 10.3390/cancers13246238

**Published:** 2021-12-12

**Authors:** Jimmy Célind, Maria Bygdell, Jari Martikainen, Johan Styrke, Jan-Erik Damber, Jenny M. Kindblom, Claes Ohlsson

**Affiliations:** 1Centre for Bone and Arthritis Research, Sahlgrenska Osteoporosis Centre, Institute of Medicine, Sahlgrenska Academy, University of Gothenburg, 405 30 Gothenburg, Sweden; maria.bygdell@gu.se (M.B.); jenny.kindblom@gu.se (J.M.K.); claes.ohlsson@gu.se (C.O.); 2Department of Pediatrics, Institute of Clinical Sciences, The Sahlgrenska Academy, University of Gothenburg, 405 30 Gothenburg, Sweden; 3Bioinformatics Core Facility, Sahlgrenska Academy, University of Gothenburg, 405 30 Gothenburg, Sweden; jari.martikainen@gu.se; 4Department of Surgical and Perioperative Sciences, Urology and Andrology, Umeå University, 901 87 Umeå, Sweden; johan.styrke@umu.se; 5Department of Urology, Institute of Clinical Sciences, The Sahlgrenska Academy, University of Gothenburg, 405 30 Gothenburg, Sweden; jan-erik.damber@urology.gu.se; 6Pediatric Clinical Research Center, Sahlgrenska University Hospital, Region Västra Götaland, 416 50 Gothenburg, Sweden; 7Department of Drug Treatment, Sahlgrenska University Hospital, Region Västra Götaland, 413 45 Gothenburg, Sweden

**Keywords:** prostate cancer, peak height velocity, pubertal timing, epidemiology

## Abstract

**Simple Summary:**

Men’s pubertal timing lacks distinct markers that are easily available retrospectively. Therefore, the association between objectively assessed pubertal timing and the risk of prostate cancer is unknown. Our aim was to evaluate the association between the age at the pubertal growth spurt, an objective assessment of pubertal timing, and the risk of prostate cancer and high-risk prostate cancer. We used a population-based cohort including over 30,000 men with age at the pubertal growth spurt available and with follow-up in high quality national registers. During 1.4 million years of follow up, 1759 cases of prostate cancer were diagnosed. We demonstrate that late pubertal timing is a protective factor for prostate cancer, and especially for the clinically important high-risk or metastatic prostate cancer. Identification of early life risk- and protective factors for prostate cancer could provide new opportunities to unravel the underlying biological mechanism of the origins of prostate cancer.

**Abstract:**

Previous studies of pubertal timing and the risk of prostate cancer have used self-reported markers of pubertal development, recalled in mid-life, and the results have been inconclusive. Our aim was to evaluate the age at the pubertal growth spurt, an objective marker of pubertal timing, and the risk of prostate cancer and high-risk prostate cancer. This population-based cohort study included 31,971 men with sufficient height measurements to calculate age at peak height velocity (PHV). Outcomes were accessed through national registers. Hazard ratios (HR) and 95% confidence intervals (CI) were estimated by Cox regressions with follow up starting at 20 years of age. In total, 1759 cases of prostate cancer including 449 high-risk were diagnosed during follow up. Mean follow up was 42 years (standard deviation 10.0). Compared to quintiles 2–4 (Q2–4), men in the highest age at PHV quintile (Q5) had lower risk of prostate cancer (HR 0.83, 95% CI 0.73–0.94), and of high-risk prostate cancer (0.73; 0.56–0.94). In an exploratory analysis with follow up starting at age at PHV, late pubertal timing was no longer associated with reduced risk of prostate cancer. Later pubertal timing was associated with reduced risk of prostate cancer and especially high-risk prostate cancer. We propose that the risk of prostate cancer might be influenced by the number of years with exposure to adult levels of sex steroids.

## 1. Introduction

Prostate cancer is the most frequent malignancy in men in both Europe and the USA, with 248,000 new cases and 34,000 deaths estimated in the USA in 2021 [1,2]. The prevalence of prostate cancer varies largely in different regions of the world. It is most common among black men in the USA and the Caribbean, followed by white men in the USA and the Scandinavian countries, while it is uncommon in the southeast of Asia [3]. Both genetical and environmental factors contributes to the significant geographical variations. Overweight, high consumption of red meat, animal fat, and dairy products are environmental risk factors, while consumption of certain vegetables are protective factors [4,5,6]. The heterogeneity of manifest prostate cancer is also considerable. A majority of tumors are slow growing and confined to the prostate gland while 10–15% of tumors are metastasized already at diagnosis [7]. Given that prostate cancer may develop silently for many years [8], early life risk factors and accumulation of risk across the life course are plausible to be of importance.

Previous observational studies evaluating the association between pubertal timing and the risk of prostate cancer have used various self-reported markers of pubertal development, recalled in mid-life, but the results have been inconsistent [9,10,11,12,13]. Two studies did not find any significant associations between self-reported pubertal timing and risk of prostate cancer; one used self-reported growth spurt as marker of pubertal timing [9], and one used self-reported timing of shaving debut [11]. A study from California that explored self-reported age at shaving debut found a significantly increased risk of prostate cancer in the sub-group of non-white men who had shaving debut <15 years of age [13]. Two Australian studies used either self-reported growth spurt [12], or a combination-score of several self-reported pubertal markers [10]. Both found that self-reported pubertal timing later than peers reduced the risk of prostate cancer, and in one of the studies the risk of high-grade prostate cancer was also reduced [12].

Puberty initiates the exposure to adult sex steroid levels. In girls, age at menarche has substantial accuracy, even recalled in midlife, but the accuracy of recalled male pubertal markers has not been established [14]. Therefore, studies on how male pubertal timing may influence the risk of disease are scarce. If several measurements of height during pubertal growth are available, age at the peak height velocity (PHV) can be estimated and used as an objective assessment of pubertal timing [15]. PHV is a mid-pubertal marker [16], and although there are individual differences in pubertal trajectories, age at PHV on a group level has been shown to associate strongly with pubertal staging according to Tanner, including genital and testicular growth, and pubic hair development [17].

In this study we hypothesized that late pubertal timing and thereby reduced number of years with exposure to adult levels of sex steroids is associated with lower risk of prostate cancer. The population-based BMI Epidemiology Study (BEST) Gothenburg with age at PHV and detailed information on prostate cancer and prostate cancer severity available for a large cohort of men, provides a unique setting to test this hypothesis.

## 2. Patients and Methods

### 2.1. Study Population

In this population-based cohort study, we collected birthweight as well as height and weight measurements from centrally archived School Health Care records for all men born between 1945 and 1961 who finished school in Gothenburg, Sweden. We also collected height and weight data for the included subjects from military conscription tests. Conscription was mandatory for all Swedish men until 2010. The study cohort was linked to national registers using the personal identity numbers (PINs). Moreover, we retrieved heights from the passport register, which includes self-reported heights for all individuals holding a passport in Sweden. The details of the BEST cohort have been described previously [18]. Eligible individuals in the present study were those with a School Health Care record in the central archive and a ten-digit PIN. The inclusion process for the present study (*N* = 31,971) is shown in Figure 1. The men in the study were followed from 20 years of age until censoring due to a prostate cancer diagnosis, death, migration, or until 31 December 2019, whichever came first. Individuals who migrated or died before the age of 20 were excluded. The Ethics Committee of the University of Gothenburg, Sweden approved the study and waived the need for informed consent. There was no commercial sponsorship.

### 2.2. Exposures

Our main exposure was age at PHV, defined as the age at the maximum growth velocity during puberty. Age at PHV was estimated according to a modified infancy-childhood-puberty model which fits human growth data to mathematical functions [15]. Estimation of age at PHV requires height measurements before, during, and after the pubertal period. Height was measured in a standardized manner by trained personal using wall-mounted stadiometers in School Health Care and at conscription. For each individual growth curve with sufficient information on height, the Infancy-Childhood-Puberty model was fitted by minimizing the sum of squares using a modification of the Levenberg–Marquardt algorithm [15] as previously described [19]. Covariates BMI at 8 and 20 years of age, together with height at 8 years and final height, are described in the Appendix A.

### 2.3. Outcomes

Information on outcomes were obtained from the National Prostate Cancer Register (NPCR) which captures 98 % of diagnosed prostate cancers in Sweden [20]. The dates and diagnoses for prostate cancer together with information on prognostic factors were retrieved for the individuals in the study. Classification of prostate cancers into 1, low or intermediate risk; 2, high-risk or metastatic prostate cancer was based on PSA-level, Gleason-score, and TNM-stage at diagnosis (Table 1). In addition, we also included individuals without prior registration in the NPCR who died due to prostate cancer (as underlying cause of death) from the Swedish Cause of Death Register (*N* = 4).

### 2.4. Statistical Analyses

Differences between quintile one (Q1) and Q5 of age at PHV compared to reference Q2–4 were tested using chi square test for dichotomous variables and Student’s t-test for continuous variables (Appendix A). Hazard ratios (HR) and 95% Confidence Intervals (CI) were estimated using a Cox regression model adjusted for birth year, and country of birth. The assumption of proportionality was confirmed for the association between age at PHV and the risk of any prostate cancer and high-risk prostate cancer. We further adjusted the analyses for BMI at 8 and 20 years of age, height at 8 years and final height, birthweight, and level of education. Non-linear associations were tested by inclusion of a quadratic term for the variable of interest in the cox regression model. A *p*-value less than 0.05 was considered statistically significant. The number of cases per 100,000 follow up years in the compared groups were calculated. Possible interactions were tested by inclusion of an interaction term in the cox regression models. The interaction term was the product of the two parameters (continuous) of interest. Kaplan–Meier analyses were performed with follow up starting at 20 years of age, and as an exploratory analysis with follow up starting at age at PHV (Figure 2). A statistical difference between groups in the Kaplan–Meier analyses was tested using a log-rank test. The statistical analyses were performed in R or SPSS version 27. Cumulative incidence plots were used to evaluate potential competing risks. As a sensitivity analysis, high or low-risk prostate cancer were categorized solely on the Gleason score at diagnosis (Appendix A).

## 3. Results

A total of 31,971 individuals met the inclusion criteria (Figure 1). The mean age at PHV was 14.1 years, and the standard deviation (SD) 1.1 (Appendix A). During the 1.4 million person-years of follow up, altogether 1759 men were diagnosed with prostate cancer and among them, 449 with high-risk or metastatic prostate cancer and 1310 with low or intermediate risk prostate cancer (see definition in Table 1). The mean age at prostate cancer diagnosis was 61.6 years (SD 5.4 years), and for high-risk or metastatic prostate cancer 63.0 (5.4).

Age at Peak Height Velocity and Risk of Prostate Cancer

Age at PHV showed a non-linear association with the risk of prostate cancer (*p* < 0.01 for (age at PHV)^2^). We evaluated the distribution of prostate cancer cases over the range of age at PHV, and found that there were significantly fewer cancer cases among the 20% with the latest age at PHV (the quintile with the latest age at PHV, Q5, with age at PHV >15 years of age) compared with the middle three quintiles (Q2–4) (*p* < 0.05) (Appendix A). Individuals in Q5 had a significantly lower risk of prostate cancer compared with the reference quintiles Q2–4 (HR 0.83 (95% CI 0.73–0.94); Table 2, Figure 2A). The reduced risk represented 15 fewer cases of prostate cancer per 100.000 follow up years for individuals with late pubertal timing (Q5) compared to average (Q2–4) (Table 3). In contrast, there was no significant association for the earliest 20% of age at PHV (Q1) with the risk of prostate cancer, compared with Q2–4 (Table 2).

Cox proportional hazards regression of the association between pubertal timing and prostate cancer (all cases n = 1759), also stratified in high-risk or metastatic (n = 449) prostate cancer, and low or intermediate-risk (n = 1310) prostate cancer at diagnosis. Analyses were adjusted for birth year and country of birth. Total *N* = 31,971. Each quintile represents 6394 individuals except for Q3 that represents 6395 individuals. High-risk or metastatic prostate cancer was defined as any of PSA >20 ng/mL, Gleason score 4 + 3 = 7 or higher, T3–4, N1 or M1.

Next, we evaluated whether age at PHV associated with lower risk of the clinically important group of high-risk or metastatic prostate cancer (see Table 1 for definition). We found that individuals with age at PHV in Q5 had substantially reduced risk of high-risk or metastatic prostate cancer (HR 0.73, 95% CI 0.56–0.94) compared with Q2–4 (Table 2, Figure 2C). The group with prostate cancer not fulfilling the definition of high-risk or metastatic prostate cancer consists of both low and intermediate risk tumors. Individuals in Q5 had moderately decreased risk of low and intermediate risk prostate cancer, compared with Q2–4 (Table 2).

Of note, among the 1759 prostate cancer cases, men in Q5 had substantially lower (−29%) risk of high-risk or metastatic prostate cancer compared with men in Q2–4 (HR 0.71, 95% CI 0.55–0.92), demonstrating that late puberty not only associates with reduced overall risk of prostate cancer but also with a better prognosis of the diagnosed prostate cancer.

Moreover, in an exploratory analysis we let the follow up start at each individual’s age at PHV instead of at age 20 years. Interestingly, the lower risk of prostate cancer, and of high-risk or metastatic prostate cancer seen for Q5 compared with Q2–4 was completely lost in these analyses (Figure 2B,D). There was no association between Q5 of age at PHV and the risk of prostate cancer (HR 0.99, 95%CI 0.87–1.12; Figure 2B) or high risk or metastatic prostate cancer (HR 0.92, 95% CI 0.71–1.20, Figure 2D) compared with Q2–4 when the follow up started at age at PHV. These findings indicate that the observed associations are driven by the duration of exposure to adult sex steroid levels after puberty.

These findings were robust the adjustment for additional covariates (Appendix A). Sensitivity analyses categorizing prostate cancer into high or low/intermediate-risk based on Gleason score alone showed similar results as the main analyses, although the result for early pubertal timing (Q1) showed a non-significant tendency toward increased risk for high-risk prostate cancer (Appendix A). For further adjusted analyses and analyses of competing risk, see the Appendix A.

## 4. Discussion

Studies of the association between pubertal timing and the risk of prostate cancer are scarce due to the dearth of distinct markers of pubertal timing in men. In the present study we used an objective assessment of pubertal timing for over 30,000 men in a population-based cohort. We demonstrate lower risk of overall prostate cancer and especially of high-risk or metastatic prostate cancer among individuals with late pubertal timing, compared with individuals with average pubertal timing. No significant association was seen for those with early pubertal timing. In addition, among men with prostate cancer, late pubertal timing was associated with prostate cancer with better prognosis.

Previous studies evaluating the association between pubertal timing and the risk of prostate cancer have used self-reported timing of pubertal markers such as shaving initiation, voice break or pubertal growth spurt, recalled in mid-life several decades after puberty, with inconclusive results [9,10,11,12,13]. The results in the present study are consistent with the previous studies showing lower risk for prostate cancer in individuals that report a growth spurt later than peers [12], and a later pubertal timing based on a combination of pubertal factors [10]. The present results are also in line with the Finnish study that showed a non-significant tendency toward a protective effect from self-reported late pubertal timing [9]. While the pubertal markers represent distinct indicators of puberty, their retrospective use has not been validated and may be limited due to recall bias, as demonstrated in a study from Harvard where only 8% of males in their 50s remembered the year of their growth spurt correctly [25]. In the present study, we used the age at which the maximum longitudinal growth velocity was attained (age at peak height velocity) as a marker of pubertal timing [15,19]. Age at PHV has been shown to associate strongly with pubertal timing according to Tanner, retrieved from detailed longitudinal physical examinations of secondary sex characteristics [17]. Age at PHV thus represents an objective accurate method to determine pubertal timing retrospectively.

The findings in the present well-powered observational study are in agreement with a Mendelian Randomization (MR) study, using genetic instruments mainly developed for menarche in girls, suggesting that later pubertal timing is boys is causally associated with reduced risk of adult prostate cancer [26].

The mechanism behind the association between pubertal timing and endocrine-related female cancers is not fully known, but the importance of the duration of exposure to adult sex steroid levels has been discussed as one plausible contributing aspect [27]. It is well-established that androgen stimulation is important for growth and survival of prostate cancer tumors [28], but the relation between serum testosterone levels and prostate cancer has been debated [29]. A causal association between serum levels of bioavailable testosterone and the risk of prostate cancer was recently demonstrated using the MR approach [30], and a protective effect of low levels of bioavailable testosterone has also been proposed [31]. Interestingly, when we let the follow up start at the pubertal growth spurt of each individual, the difference regarding the risk of prostate cancer between individuals with late and average pubertal timing was lost. The observational nature of our study precludes making conclusive mechanistic statements about the observed associations, but our findings can be useful for hypothesis generation. We propose that the risk of overall prostate cancer and especially the more severe high-risk or metastatic prostate cancer might be influenced by the number of years with exposure to adult levels of sex steroids (=the number of years after puberty). We detected a non-linear association between pubertal timing and prostate cancer in the present study, where late pubertal timing was associated with reduced risk while early pubertal timing was not associated with increased risk of prostate cancer, compared with individuals with average pubertal timing. The exact pathophysiological mechanism behind this is not known, but one may speculate that the sensitivity to testosterone action is different in early/average compared to late pubertal timing and that this might contribute to the non-linear association. 

It is plausible that both the adult serum levels of bioavailable testosterone, as indicated by the recent MR study [30], and the duration of exposure to adult sex steroid levels (as indicated by the present observational study and a MR study [26]) influence the risk of prostate cancer.

Given the large heterogeneity in prognosis within the prostate cancer diagnosis [28], it is of importance to evaluate the association specifically with high-risk or metastatic tumors known to confer a worse prognosis and increased risk of mortality [21]. Previous studies, both observational [9] and using the MR approach [26], defined high-risk as Gleason score from 7 and above. This results in tumors of intermediate risk (with Gleason grades 3 + 4) being included as high-risk [21], which is not in line with the risk categorization in current international guidelines [21]. In the present study, we used a clinically relevant definition of high-risk tumors including Gleason grades 4+3 and above, together with PSA levels and TNM-stage to define high-risk tumors according to current guidelines [23,32]. Using this definition, we demonstrate an association with reduced risk especially of high-risk or metastatic prostate cancer for individuals with late pubertal timing, and we also demonstrate that among men with prostate cancer, late pubertal timing was associated with prostate cancer with better prognosis. The clinical relevance of the present study includes that late pubertal timing is a plausible novel protective factor for prostate cancer, and highlights that risk of prostate cancer may start to develop already in early life.

The strengths of the present study include the population-based design, the large and well-powered cohort, the long and near complete follow up in Swedish high-quality registers [20,33,34] and with detailed information from the National Prostate Cancer Register [20] enabling us to categorize prostate cancer into high risk and metastatic / low and intermediate prostate cancer. In addition, we used an objective accurate estimation of pubertal timing. An important limitation is that the Swedish population at the time of our cohort´s school attendance was primarily Caucasian, which is why generalizability to other ethnicities is limited. The mean age at diagnosis of prostate cancer in Sweden is 70 years [35], and the mean age at end of follow up in the present cohort is 62 years. Therefore, the results from the present study mainly reflects early prostate cancer. Moreover, we cannot rule out residual confounding of the results.

In conclusion, we demonstrate that objectively measured late pubertal timing is associated with lower risk of overall prostate cancer and especially of high-risk or metastatic prostate cancer. We propose that the risk of overall prostate cancer and especially the more severe high-risk or metastatic prostate cancer might be influenced by the number of years with exposure to adult levels of sex steroids.

## Figures and Tables

**Figure 1 cancers-13-06238-f001:**
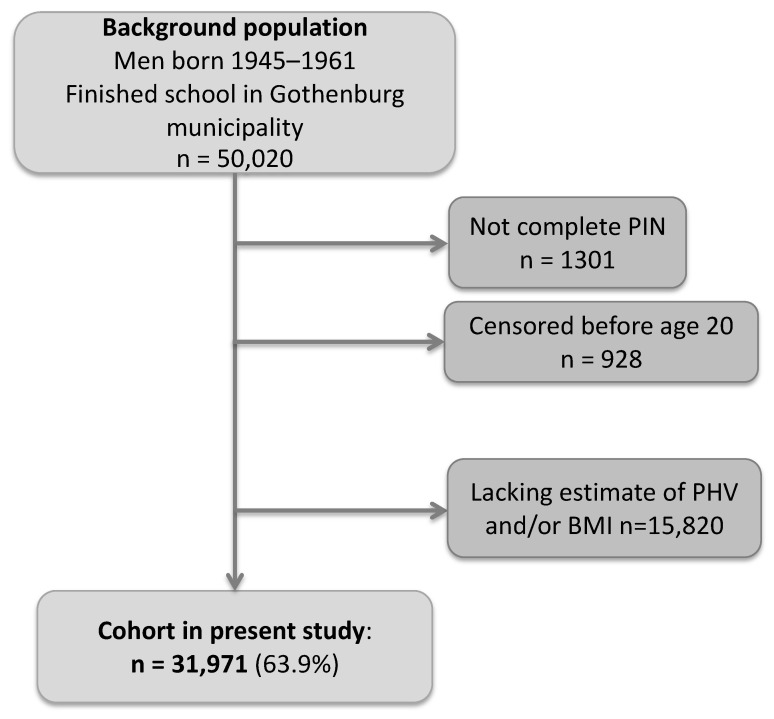
Flow chart of the inclusion process. BMI = Body Mass Index. PIN = Personal Identity Number.

**Figure 2 cancers-13-06238-f002:**
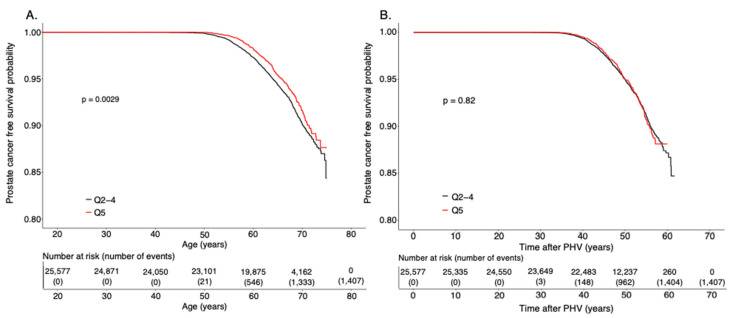
(**A**–**D**) Kaplan–Meier survival curves from prostate cancer or high-risk or metastatic cancer. Quintile 5 (Q5) compared to Q2–4 for survival from any prostate cancer (**A**,**B**) or high-risk or metastatic prostate cancer (**C**,**D**) with follow up starting at 20 years of age (**A**,**C**), or at age at Peak Height Velocity (PHV) (**B**,**D**).

**Table 1 cancers-13-06238-t001:** Classification of high-risk or metastatic prostate cancer.

Risk Classification at Diagnosis	Definition
High-risk or metastatic	Any of: PSA > 20 ng/mL Gleason score 4 + 3 = 7 or higher Tumor stage T3–4 Regional lymph node metastasis N1 Metastasis M1
Low or intermediate risk	All prostate cancers not categorized as high-risk

Classification of high-risk prostate cancer was based on the International Society for Urologist Pathology Consensus [21], the classical definition by D’Amico [22], the STAR-CAP cohort [23], and the Swedish National Guidelines for prostate cancer [24].

**Table 2 cancers-13-06238-t002:** Quintiles of age at Peak Height Velocity and risk of prostate cancer.

	HR (95% CI) for Age at PHV		
Quintiles	Prostate cancer	High-risk or metastatic prostate cancer	Low or intermediate risk prostate cancer
Q1	0.94 (0.83–1.06)	0.86 (0.67–1.09)	0.97 (0.84–1.11)
Q2–4	Ref	Ref	Ref
Q5	0.83 (0.73–0.94)	0.73 (0.56–0.94)	0.86 (0.74–0.99)

**Table 3 cancers-13-06238-t003:** Cases of prostate cancer and high-risk or metastatic prostate cancer per 100,000 years of follow up for late compared to average pubertal timing.

Quintiles of Age at PHV	Number of Cases of Prostate Cancer/100,000 Follow Up Years
Q2–4	92
Q5	77
Difference	15

Quintile 2–4 (Q2–4) are men with an average pubertal timing and quintile 5 are those in the latest fifth of pubertal timing. Pubertal timing was assessed through the age at peak height velocity (PHV). Each quintile represents 6394 individuals except for Q3 that represents 6395 individuals. Total *N* = 31,971.

## Data Availability

Research data are not publicly available due to privacy and ethical restrictions. However, anonymized data that are minimally required to reproduce results can be made available from the corresponding author upon reasonable request, upon approval from the University of Gothenburg, if the data can be made available according to mandatory national law.

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
