# Peer review of "Timing of the Pubertal Growth Spurt and Prostate Cancer"

_cancers, 2021, doi:10.3390/cancers13246238_

Round 1

Reviewer 1 Report

Aim of the current paper was to investigate any association between duration of exposure to sex hormones, considering age at peak height velocity as a marker of pubertal timing, and the risk of developing prostate cancer.

The manuscript is well written and the topic is of major importance. References are accurate. However, there are several issues which should be better addressed:

  • Although this study has showed an association between a reduced risk in developing prostate cancer and late pubertal timing, there was no evidence that an early pubertal timing could have led to increased risk in developing prostate cancer. As a matter of fact, there was no significant association between an early age at PHV and the risk of prostate cancer. This should be more extensively discussed. Such observation lets me believe that there may be some additional confounding factors (as in all those cases in which an attempt is made to identify a single etiopathogenetic factor for multifactorial disorders)
  • In addition, this study assumes that the first exposure to increased testosterone levels occurs at the peak height velocity, although the testosterone levels begin to increase almost a year before the growth spurt, corresponding to male genitalia growth. Such observation might imply that there are some men grouped in the same quintile who could have been exposed longer than others to the influence of sex hormones.
  • The different features of the reference group (Q2-Q4) have not clearly been specified.

Author Response

Reviewer 1

  1. Although this study has showed an association between a reduced risk in developing prostate cancer and late pubertal timing, there was no evidence that an early pubertal timing could have led to increased risk in developing prostate cancer. As a matter of fact, there was no significant association between an early age at PHV and the risk of prostate cancer. This should be more extensively discussed. Such observation lets me believe that there may be some additional confounding factors (as in all those cases in which an attempt is made to identify a single etiopathogenetic factor for multifactorial disorders)

Response: This Reviewer is correct in that we do not see an increased risk in individuals with early pubertal timing (age at peak height velocity). Instead, we found a non-linear association where the risk was lower for those with late pubertal timing. We have emphasized the non-linear nature of the association observed, and as suggested we also added a more extensive discussion about the notion that early pubertal timing was not associated with increased risk of prostate cancer. In addition, we mention the possibility of residual confounding as a limitation.

Discussion page 7, 5th paragraph: No significant association was seen for those with early pubertal timing.

Discussion page 8, 1st paragraph: We detected a non-linear association between pubertal timing and prostate cancer in the present study, where late pubertal timing was associated with reduced risk while early pubertal timing was not associated with increased risk of prostate cancer, compared with individuals with average pubertal timing. The exact pathophysiological mechanism behind this is not known, but one may speculate that the sensitivity to testosterone action is different in early/average compared to late pubertal timing and that this might contribute to the non-linear association. 

Discussion page 8, 4th paragraph: Moreover, we cannot rule out residual confounding of the results. 

  1. In addition, this study assumes that the first exposure to increased testosterone levels occurs at the peak height velocity, although the testosterone levels begin to increase almost a year before the growth spurt, corresponding to male genitalia growth. Such observation might imply that there are some men grouped in the same quintile who could have been exposed longer than others to the influence of sex hormones.

Response: We completely agree that peak height velocity does not indicate the first exposure of testosterone in an individual. However, the age at peak height velocity correlates strongly with Tanner stages, the gold standard for pubertal staging in the clinic including genital and testicular growth, and also pubic hair development (1). This indicates that on a group level, age at PHV is a valid measure of pubertal timing. Given the scarcity of male population-based cohorts with repeated genital examinations, and the poor validity of self-reported pubertal timing in males (2), age at peak height velocity is often the only valid retrospective indicator of pubertal timing. We have made clarifications regarding PHV and its validity as a pubertal marker.

Introduction page 2, 7th paragraph: PHV is a mid-pubertal marker (3), and although there are individual differences in pubertal trajectories, age at PHV on a group level has been shown to associate strongly with pubertal staging according to Tanner, including genital and testicular growth, and pubic hair development (1).

  1. The different features of the reference group (Q2-Q4) have not clearly been specified.

Response: An additional table has been added to the supplementary (Supplementary Table S1B) to provide more details about the reference group (Q2–Q4).

See Suppl Table S1B.

Reviewer 2 Report

The article entitled Timing of the pubertal growth spurt and prostate cancer by authors is well written with sufficient scientific data. However, presentation need to improve. In my opinion authors must be rearrange the content presented here to improve its readability and understanding purpose. Conclusion part must be improved.

After considering these points this article can be published in Cancers journal.

Author Response

Reviewer 2

The article entitled Timing of the pubertal growth spurt and prostate cancer by authors is well written with sufficient scientific data. However, presentation need to improve. In my opinion authors must be rearrange the content presented here to improve its readability and understanding purpose. Conclusion part must be improved.

Response: Thank you for the appreciative feedback that our manuscript is well written and has sufficient scientific data. We have rearranged the Introduction and Discussion parts, and also added two new Supplementary tables to clarify the results. Moreover, the Conclusion has been revised.

Conclusion, page 8, last paragraph: In conclusion, we demonstrate that objectively measured late pubertal timing is associated with lower risk of overall prostate cancer and especially of high-risk or metastatic prostate cancer. We propose that the risk of overall prostate cancer and especially the more severe high-risk or metastatic prostate cancer might be influenced by the number of years with exposure to adult levels of sex steroids.

See Supplementary tables S1B and S3

Reviewer 3 Report

  1. The manuscript could benefit from editing for grammar, missing words, and subject-verb agreement, etc. It is recommended that authors delete irrelevant "general" phrases and sentences, repeated and unneeded words. They should use short sentences. Also, some Introductory sentences are irrelevant or are not needed.
  2. Abstract: “follow up starting at 20 years of age.” Please add the mean follow up time.
  3. Abstract: in the background, I recommend authors to add some introductory information about what is known regarding pubertal age and prostate cancer.
  4. Abstract: please define the quantiles of age and their ranges.
  5. All abbreviations should be revised and defined at their first use.
  6. Introduction: “Prostate cancer is the most frequent malignancy in European men.” Reference 1 goes back to 2015. Please update. Siegel et al. 2021 cancer statistics data could be used instead for updated numbers of colon cancer incidence. The whole manuscript should be revised, references should be updated.
  7. Introduction: Introduction is short. I recommend adding some background information about the other risk factors of PCa.
  8. References: please leave a space before citing references such as: “deaths in the European Union 2015[1].” Add a space before [1]. Revise the whole manuscript and correct accordingly.
  9. Introduction: “results have been inconsistent[4-8].” Give examples.
  10. Methods: Did authors acquire IRB approval? And what about patient consent? Please add details.
  11. Methods: “Precise estimation of age at PHV requires height measurements before, during, and after the pubertal period.” How accurate were the information collected? This might represent a limitation as errors in reporting might have been present.
  12. Methods: What defines high-risk prostate cancer? Based on what criteria did authors assume that PSA > 20ng/mL is a high-risk prostate cancer patient? Same applies to pN1 and pT3-4.
  13. Methods: please define the five quantiles of age.
  14. Results: Did authors adjust results for parameters such as PSA, gleason score, pT and pN stages, etc.? Stratifying patients could have been done also by categorizing patients into low-risk (grade groups 1 and 2 = Gleason 6(3+3) and 7(3+4); and high-risk (grade groups 3, 4, and 5 = Gleason 7(4+3), 8, and 9).
  15. Discussion section: Authors should focus more on the main findings and avoid repeating results presentation in the discussion. Authors could also correlate their findings with what has been published in literature. Clinical relevance should be added.

Author Response

Reviewer 3

  1. The manuscript could benefit from editing for grammar, missing words, and subject-verb agreement, etc. It is recommended that authors delete irrelevant "general" phrases and sentences, repeated and unneeded words. They should use short sentences. Also, some Introductory sentences are irrelevant or are not needed.

Response: The manuscript has been revised, grammar has been corrected, unnecessary phrases have been removed, and long sentences have been made shorter. See tracked changes throughout the manuscript.

  1. Abstract: “follow up starting at 20 years of age.” Please add the mean follow up time.

Response: The mean follow up time was added to the abstract.

Abstract page 2, 2nd paragraph: Mean follow up was 42 years (standard deviation 10.0).

  1. Abstract: in the background, I recommend authors to add some introductory information about what is known regarding pubertal age and prostate cancer.

Response: We have added information regarding pubertal timing and prostate cancer to the abstract background.

Abstract page 1, 1st paragraph: Previous studies of pubertal timing and the risk of prostate cancer have used self-reported markers of pubertal development, recalled in mid-life, and the results have been inconclusive.

  1. Abstract: please define the quantiles of age and their ranges.

Response: Detailed information regarding the quintiles have been added in a new table, Suppl Table S1B.

See Suppl Table S1B.

  1. All abbreviations should be revised and defined at their first use.

Response: The manuscript has been revised and abbreviations are defined at their first use.

  1. Introduction: “Prostate cancer is the most frequent malignancy in European men.” Reference 1 goes back to 2015. Please update. Siegel et al. 2021 cancer statistics data could be used instead for updated numbers of colon cancer incidence. The whole manuscript should be revised, references should be updated.

Response: We thank this reviewer for the recommended reference. The references have been updated accordingly.

  1. Introduction: Introduction is short. I recommend adding some background information about the other risk factors of PCa.

Response: We have added more background information (and references) about risk factors of prostate cancer to the Introduction.

Introduction page 2, 5th paragraph: The prevalence of prostate cancer varies largely in different regions of the world. It is most common among black men in the USA and the Caribbean, followed by white men in the USA and the Scandinavian countries, while it is uncommon in the southeast of Asia (4). Both genetical and environmental factors contributes to the significant geographical variations. Overweight, high consumption of red meat, animal fat, and dairy products are environmental risk factors, while consumption of certain vegetables are protective factors (5-7).  

  1. References: please leave a space before citing references such as: “deaths in the European Union 2015[1].” Add a space before [1]. Revise the whole manuscript and correct accordingly.

Response: A space have been inserted before citing references.

  1. Introduction: “results have been inconsistent[4-8].” Give examples

Response: A short description of previous results has been added.

Introduction page 2, 6th paragraph: Two studies did not find any significant associations between self-reported pubertal timing and risk of prostate cancer; one used self-reported growth spurt as marker of pubertal timing (11), and one used self-reported timing of shaving debut (8). A study from California that explored self-reported age at shaving debut found a significant increased risk of prostate cancer in the sub-group of non-white men who had shaving debut <15 years of age (9). Two Australian studies used either self-reported growth spurt (10), or a combination-score of several self-reported pubertal markers (12). Both found that pubertal timing later than peers reduced the risk of prostate cancer, and in one of the studies the risk of high-grade prostate cancer was also reduced (10).

  1. Methods: Did authors acquire IRB approval? And what about patient consent? Please add details.

Response: IRB approval was given by the local Ethics Committee before the start of the study, which is declared in page 3, 1st paragraph. Information that informed consent was waived by the Ethics Committee was added to the manuscript. This decision was based on the population- and register-based nature of the study.

Methods page 3, 2nd paragraph: The Ethics Committee of the University of Gothenburg, Sweden approved the study and waived the need for informed consent. There was no commercial sponsorship.

  1. Methods: “Precise estimation of age at PHV requires height measurements before, during, and after the pubertal period.” How accurate were the information collected? This might represent a limitation as errors in reporting might have been present.

Response: The information was collected by trained professional school health care nurses and the measurements performed in a standardized manner with the pupils in light clothing, using wall-mounted stadiometers.  The model used for growth curve fitting in the estimation of age at PHV has been described before and the research group has extensive experience using it (13, 14). We added this information and removed the word “precise” from the description of the method.

Methods page 4, 1st paragraph: Height was measured in a standardized manner by trained personal using wall-mounted stadiometers in School Health Care and at conscription.

  1. Methods: What defines high-risk prostate cancer? Based on what criteria did authors assume that PSA > 20ng/mL is a high-risk prostate cancer patient? Same applies to pN1 and pT3-4.

Response: The definition of high-risk prostate cancer was based on the STAR-CAP cohort (15), the classical definition by D’Amico (16), the consensus criteria from the International Society of Urological Pathology (17), and the Swedish National Guidelines for Prostate Cancer (18). This has been clarified in the manuscript.

Page 4, Table 1: Classification of high-risk prostate cancer was based on the International Society for Urologist Pathology Consensus (17), the classical definition by D’Amico (16), the STAR-CAP cohort (15), and the Swedish National Guidelines for prostate cancer (18).

  1. Methods: please define the five quantiles of age.

Response: See response to comment no 4.

  1. Results: Did authors adjust results for parameters such as PSA, gleason score, pT and pN stages, etc.? Stratifying patients could have been done also by categorizing patients into low-risk (grade groups 1 and 2 = Gleason 6(3+3) and 7(3+4); and high-risk (grade groups 3, 4, and 5 = Gleason 7(4+3), 8, and 9).

Response: PSA, gleason score, pT and pN stages are all parameters included in the outcome, and therefore they have not been used as covariates. However, we did perform sensitivity analyses for stratification into high or low risk prostate cancer based on Gleason score alone. These results have been added to the manuscript together with a new table, Supplementary Table S3.

Results page 7, 4th paragraph: Sensitivity analyses categorizing prostate cancer into high or low/intermediate-risk based on Gleason score alone showed similar results as the main analyses, although the result for early pubertal timing (Q1) showed a non-significant tendency toward increased risk for high-risk prostate cancer (Supplementary Table S3).

  1. Discussion section: Authors should focus more on the main findings and avoid repeating results presentation in the discussion. Authors could also correlate their findings with what has been published in literature. Clinical relevance should be added.

Response: The Discussion has been revised to avoid repeating results, and correlation with previous literature has been extended. A section relating to clinical relevance has been added.

Discussion page 7, 6th paragraph: The results in the present study are consistent with the previous studies showing lower risk for prostate cancer in individuals that reported of a growth spurt later than peers (10), and a later pubertal timing based on a combination of pubertal factors (12). The present results are also in line with the Finnish study that showed a non-significant tendency toward a protective effect from self-reported late pubertal timing (11).

Discussion page 8, 3rd paragraph: The clinical relevance of the present study includes that late pubertal timing is a plausible novel protective factor for prostate cancer, and highlights that risk of prostate cancer may develop already in early life.